# Characterization of Adherent-Invasive *Escherichia coli* (AIEC) Outer Membrane Proteins Provides Potential Molecular Markers to Screen Putative AIEC Strains

**DOI:** 10.3390/ijms23169005

**Published:** 2022-08-12

**Authors:** Waleska Saitz, David A. Montero, Mirka Pardo, Daniela Araya, Marjorie De la Fuente, Marcela A. Hermoso, Mauricio J. Farfán, Daniel Ginard, Ramon Rosselló-Móra, Dave A. Rasko, Felipe Del Canto, Roberto M. Vidal

**Affiliations:** 1Programa de Microbiología y Micología, Instituto de Ciencias Biomédicas, Facultad de Medicina, Universidad de Chile, Santiago 8380453, Chile; 2Programa de Inmunología, Instituto de Ciencias Biomédicas, Facultad de Medicina, Universidad de Chile, Santiago 8380453, Chile; 3Centro Integrativo de Biología y Química Aplicada (CIBQA), Universidad Bernardo O’Higgins, Santiago 8370993, Chile; 4Department of Gastroenterology and Hepatology, University Medical Center Groningen (UMCG), University of Groningen, 9712 Groningen, The Netherlands; 5Departamento de Pediatría y Cirugía Infantil Oriente, Hospital Dr. Luis Calvo Mackenna, Facultad de Medicina, Universidad de Chile, Santiago 7500539, Chile; 6Department of Gastroenterology and Palma Health Research Institute, Hospital Universitario Son Espases, 07120 Palma de Mallorca, Spain; 7Grupo de Microbiología Marina, Instituto Mediterráneo de Estudios Avanzados (IMEDEA; CSIC-UIB), 07190 Esporles, Illes Balears, Spain; 8Institute for Genome Sciences, University of Maryland School of Medicine, Baltimore, MD 21201, USA; 9Department of Microbiology and Immunology, University of Maryland School of Medicine, Baltimore, MD 21201, USA; 10Instituto Milenio de Inmunología e Inmunoterapia, Facultad de Medicina, Universidad de Chile, Santiago 8380453, Chile

**Keywords:** adherent-invasive *Escherichia coli*, AIEC, molecular marker, Crohn’s disease

## Abstract

Adherent-invasive *E. coli* (AIEC) is a pathotype associated with the etiopathogenesis of Crohn’s disease (CD), albeit with an as-yet unclear role. The main pathogenic mechanisms described for AIEC are adherence to epithelial cells, invasion of epithelial cells, and survival and replication within macrophages. A few virulence factors have been described as participating directly in these phenotypes, most of which have been evaluated only in AIEC reference strains. To date, no molecular markers have been identified that can differentiate AIEC from other *E. coli* pathotypes, so these strains are currently identified based on the phenotypic characterization of their pathogenic mechanisms. The identification of putative AIEC molecular markers could be beneficial not only from the diagnostic point of view but could also help in better understanding the determinants of AIEC pathogenicity. The objective of this study was to identify molecular markers that contribute to the screening of AIEC strains. For this, we characterized outer membrane protein (OMP) profiles in a group of AIEC strains and compared them with the commensal *E. coli* HS strain. Notably, we found a set of OMPs that were present in the AIEC strains but absent in the HS strain. Moreover, we developed a PCR assay and performed phylogenomic analyses to determine the frequency and distribution of the genes coding for these OMPs in a larger collection of AIEC and other *E. coli* strains. As result, it was found that three genes (*chuA*, *eefC*, and *fitA*) are widely distributed and significantly correlated with AIEC strains, whereas they are infrequent in commensal and diarrheagenic *E. coli* strains (DEC). Additional studies are needed to validate these markers in diverse strain collections from different geographical regions, as well as investigate their possible role in AIEC pathogenicity.

## 1. Introduction

Adherent-invasive *E. coli* (AIEC) strains were first isolated from the ileal mucosa of patients with Crohn’s disease (CD) and defined as a new pathotype due to their ability to adhere to and invade intestinal epithelial cells and survive and replicate within macrophages, inducing high levels of TNF alpha [1,2,3,4].

The role of AIEC in the onset or chronicity of this disease is not well-defined. However, it has been proposed that these bacteria could trigger the onset of the inflammatory process as a result of the invasion of intestinal epithelial cells, and then, due to their survival within macrophages, they could promote a constant antigenic stimulus, chronic inflammation, and the development of granulomas [4]. Interestingly, AIEC strains have also been isolated from other mammals (dogs, cats, and cattle) with cases of granulomatous colitis and bovine mastitis, demonstrating their proinflammatory activity [5,6,7].

The adherence and invasion phenotypes are not sufficient to distinguish AIEC from other *E. coli* pathotypes. However, the invasion capacity of AIEC is not associated with the classical genetic determinants of invasion present in diarrheagenic *E. coli* strains (DEC), such as the *ipaC* and *tia* genes that encode for invasins of enteroinvasive *E. coli* and enterotoxigenic *E. coli*, respectively [3,8].

AIEC strains can survive and replicate within macrophages, including the murine cell line J774-A1, murine peritoneal macrophages, and human monocyte-derived macrophages. When phagocytosed, AIEC strains induce the fusion of phagosomes in order to develop a large vacuole where they can survive and replicate for a long period of time. This process occurs without causing cell death or apparent damage to phagocytic cells, but it does trigger the secretion of high levels of tumor necrosis factor α (TNF-α) [2]. This differentiates AIEC from other *E. coli* pathotypes, which have cytotoxic effects on macrophages [9].

To date, several studies have been conducted to identify virulence factors of AIEC and to better understand its pathogenicity. These factors include flagellum and type I pili, which are required to adhere to epithelial cells [10,11]; Lpf, involved in its translocation across M cells [12]; and IbeA, OmpA, and outer membrane vesicles (OMVs), which are necessary to invade cells [13]. There is little knowledge regarding the mechanism used by AIEC to survive and replicate inside macrophages. Although a correlation of the presence of *chuA* (heme acquisition) with AIEC persistence in macrophages has been reported [14], there are no studies that confirm its role in AIEC pathogenicity. 

Many comparative genomic studies of AIEC strains from different collections have been carried out to identify molecular markers; however, no specific marker capable of distinguishing the AIEC pathotype has been found [15,16]. These studies have supported the idea that AIEC does not harbor any specific genetic trait. In fact, AIEC comprises a heterogeneous group of strains that belong to several phylogenetic groups (A, B1, B2, D, and others), carrying many virulence-associated genes of extraintestinal pathogenic *E. coli* (ExPEC) strains [15,17,18]. In line with these studies, it is possible that AIEC evolved through the acquisition of a diverse set of genes, which converged synergistically to lead to the same pathogenic phenotype.

On the other hand, comparative transcriptomic studies of AIEC strains growing under particular conditions (e.g., bile salts) have shown that the differential gene expression program of these strains may drive their pathogenic phenotype [19,20]. Thus, a plausible hypothesis is that commensal *E. coli* strains and AIEC have similar gene content but differ in how these genes are expressed in the human gut environment. This makes the search for putative AIEC markers even more challenging.

Outer membrane proteins (OMPs) are exposed to the outside of the bacterial cell, and they are the first line of contact between bacteria and their environment. Given their surface location, OMPs play a critical role in diverse cellular processes, such as nutrient uptake, cell signaling, waste export, pathogenicity, and evasion of the host’s defenses [21,22]. Thus, the comprehensive analysis of the outer membrane proteome of AIEC could contribute to a better understanding of the mechanism of host–pathogen interaction and offer the opportunity to identify key virulence factors and potential therapeutic targets.

In this study, we aimed to identify OMPs associated with AIEC strains and explore their potential use as molecular markers for this pathotype. By using proteomic and genomic analyses, we identified three genes coding for OMPs, which are expressed in the presence of bile salts (an environmental signal that upregulates genes associated with AIEC colonization) and widely distributed and significantly correlated with AIEC strains. In contrast, these genes are infrequent in our collection of commensal and DEC strains. Two of these genes (*chuA* and *fitA*) are suggested to be related with iron uptake, an essential element for growth of AIEC [14]. The third gene (*eefC*) encodes a predicted multidrug efflux protein with unknown function that belongs to the TolC family. Our results suggest that these genes may be useful in the molecular characterization of AIEC. Additional studies could use external strain collections to validate these genes as markers to screen putative AIEC strains.

## 2. Materials and Methods

### 2.1. Bacterial Strains

A set of 16 *E. coli* strains isolated from patients with Crohn’s disease were included in this study (strains were kindly provided by Dr. Ramón Rosselló-Móra, Mediterranean Institute for Advanced Studies in Mallorca—CSIC, Spain) (Appendix A) [23]. The protocol and informed consent forms were approved by the Balearic Islands’ Ethical Committee, Spain. All records and information were kept confidential, and all identifiers were removed prior to analysis (certificate is enclosed). All these strains were obtained through intestinal biopsies and tissue samples subjected to treatment with 100 µg/mL of gentamicin and later with Triton-X-100/PBS to release intracellular bacteria. These strains were analyzed with pulsed-field gel electrophoresis (PFGE) and characterized according to their capacity to adhere to and invade Caco-2 cells and to survive inside RAW 264.7 macrophages, as described below. AIEC reference strains HM605 [24] and NRG857c [15] and non-adherent non-invasive *E. coli* strain HB101 were included as controls. The commensal *E. coli* HS [25] and eight additional commensal *E. coli* strains belonging to our collection were also included in this study. These commensal *E. coli* strains were isolated from healthy subjects and were found to lack virulence genes characteristic of diarrheagenic *E. coli* pathotypes [26].

### 2.2. Phylogenetic Analyses

Genetic relatedness among the 16 isolates obtained from patients with Crohn’s disease was determined by PFGE according to the protocol described by the PulseNet of the Center for Disease Control and Prevention [27]. The restriction endonuclease *Xba*I (Thermo Fisher Scientific, Waltham, MA, USA) was used to digest the genomic DNA and *Salmonella* Braenderup H1298 strain was used as a reference. Electrophoretic migration profiles were analyzed with the Dice coefficient with a 1.0% tolerance using GelCompar II software v5.10 (Applied Maths). A core SNP-based phylogenetic tree was built starting from genomic sequences using kSNP 3.1 [28]. For this purpose, a set of 81 genomes were considered, including representatives of reference AIEC, other pathogenic *E. coli*, and commensal *E. coli* strains. The tree was further processed with the Interactive Tree of Life tool [29]. Accession numbers are provided in Appendix A. Newly sequenced genomes were deposited at the NCBI database under the Bioproject code PRJNA840910. Furthermore, phylogroups were determined from the genomic sequences of each isolate according to the Clermont scheme using the EzClermont tool [30] in the batch mode.

### 2.3. Phenotypic Identification of AIEC Strains

AIEC strains were identified based on their pathogenic mechanisms: adherence to epithelial cells, invasion of epithelial cells, and survival within macrophages.

The abilities to adhere to and invade human colorectal adenocarcinoma (Caco-2) cells were evaluated as previously described [1,3,18]. Briefly, Caco-2 cells were grown in 24-well plates in high-glucose DMEM medium (glucose 4500 mg/l, L-glutamine 4.0 mM, and sodium pyruvate 110 mg/l) (HyClone, Logan, UT, USA) supplemented with 10% fetal bovine serum (FBS) and 1% antibiotic/antimycotic solution (Pen/Strep/Fungizone, HyClone, USA) until they reached a density of 5 × 10^5^ cells per well and then maintained in an atmosphere containing 5% CO_2_ at 37 °C. The cell culture was then washed with 1X PBS and infected with a bacterial suspension using a multiplicity of infection (MOI) of 10. To obtain this inoculum, bacteria were grown for 20 h without shaking in low-glucose DMEM medium (glucose 1000 mg/L, L-glutamine 4.0 mM, and sodium pyruvate 110 mg/L) (HyClone, UT, USA), from which an aliquot with approximately 5 × 10^6^ bacteria was taken and resuspended in high-glucose DMEM medium. The inoculum was confirmed by serial dilutions and counting of colony forming units (CFUs) on LB agar plates after 24 h of culture. Infected Caco-2 cells were incubated for 30 min at 37 °C with 5% CO_2_, then the cells were washed three times with 1X PBS to remove non-adherent bacteria. Subsequently, the cells were lysed using Triton X-100 at 0.1% for 15 min, and the number of CFUs recovered (adhered bacteria) from the lysates was determined by serial dilutions and plating on LB agar plates. The result was expressed as the percentage of adhered bacteria in relation to the initial inoculum. Each assay was performed in triplicate. Those strains that exhibited a percentage of adhesion greater than 0.8%, a value previously described for the characterization of AIEC strains [31], were considered adherent. This value is more than five times higher than that of the *E. coli* HB101 strain (~0.14%), which was used as a negative control.

Cell invasion assays were carried out in Caco-2 as previously described [32], with modifications. Briefly, infection of the Caco-2 cells was carried out in the same way as in the adhesion assay. Then, the infected cells were incubated for 3 h at 37 °C with 5% CO_2_. The cells were washed three times with PBS 1X and incubated in high-glucose DMEM medium supplemented with amikacin 100 µg/mL (Sigma, St. Louis, MO, USA) for 1 h at 37 °C with 5% CO_2_ to eliminate non-invasive bacteria. Invasive bacteria were determined by lysing Caco-2 cells with 0.1% Triton X-100 for 15 min and then counting CFUs on agar LB plates as described above. The result was expressed as the percentage of invasive bacteria recovered after amikacin treatment in relation to the initial bacterial inoculum. Each assay was performed in triplicate. Those strains that exhibited a percentage of invasion of Caco-2 cells greater than 0.1%, a value previously established for the characterization of AIEC strains [4], were considered invasive.

Survival and replication in macrophages were evaluated as described previously [2], with modifications. Briefly, RAW 264.7 cells were infected with the bacterial strains (MOI = 10) for 2 h at 37 °C. Cells were then washed three times with PBS and incubated in fresh high-glucose DMEM medium supplemented with 100 µg/mL amikacin. Intracellular bacterial content was determined at 1, 3, and 24 h post-infection (h.p.i.) at 37 °C with 5% CO_2_. For this, infected macrophages were lysed using 0.1% Triton X-100 for 25 min, and then CFUs were counted on agar LB plates as described above. The percentage of bacterial survival was expressed as the CFUs recovered after treatment with amikacin at 3 and 24 h compared to the CFUs recovered at the initial time (1 h after treatment with amikacin). The assay was performed in triplicate. The survival levels exhibited by the AIEC reference strains NRG857c and HM605 were considered cut-off values.

### 2.4. Outer Membrane Protein Extraction

OMP extracts were obtained as described previously [26], with minor modifications. Briefly, bacterial strains were grown in DMEM (glucose 1000 mg/L, L-glutamine 4.0 mM, and sodium pyruvate 110 mg/L) for 18 h. After that, 2% of bile salts (final concentration) were added to the cultures, and they were incubated for 90 min to stimulate the expression of virulence factors [33]. Bacteria were centrifuged at 9000× *g* for 10 min at 4 °C, and pellets were washed and resuspended in a final volume of 5 mL solution containing 10 mM Tris-HCl, pH 8.0, supplemented with 1 mM phenylmethylsulfonyl fluoride (PMSF; Sigma-Aldrich Co., St. Louis, MO, USA) and protease inhibitor cocktail (Calbiochem). Each suspension was sonicated on ice (40 cycles for 30 s with 30 s intervals) using the 3000 MP Ultrasonic homogenizer (Biologics Inc). The lysed cells were centrifuged at 12,000× *g* for 10 min at 4 °C, and the supernatant was treated with DNase-RNase at room temperature for 20 min and centrifuged at 12,000× *g* for 10 min at 4 °C. The cytoplasmic membrane was solubilized by incubating the supernatant with 2% *N*-lauroylsarcosine (Sarkosyl; Sigma-Aldrich Co., St. Louis, MO, USA) for 30 min at room temperature and then at 4 °C overnight. The next day, the mixture was centrifuged at 20,500× *g* for 90 min at 4 °C, and the pellet was washed with 1 mL of sterile Milli-Q water plus PMSF (1 mM) and centrifuged at 20,500× *g* for 1 h at 4 °C. Finally, the pellet was resuspended in 200 μL of sterile Milli-Q water. The protein concentration in the outer membrane fraction was measured using the Bradford method [34].

### 2.5. Two-Dimensional Polyacrylamide Gel Electrophoresis (2D-PAGE)

OMP extracts were analyzed with 2D-PAGE as previously described [26]. Each protein sample (250 μg) was mixed with DeStreak rehydration solution (GE Healthcare) containing 1% buffer IPG (pH 4–7) and 10 mM dithiothreitol (DTT). Each sample was loaded onto isoelectric focusing (IEF) strips (precast IPG strips, 13 cm, pH 4–7, GE Healthcare) and incubated for 16 h at room temperature. IEF was carried out in an Ettan IPGphor3 System (GE Healthcare) at 20 °C with a current limit of 50 µA/strip, according to the following program: 200 V for 1 h, 500 V for 1 h, linear gradient to 1000 V over 1 h, linear gradient to 8000 V over 3.5 h, and 8000 V for a total of 20 kV h for the entire run. After focusing, the strips were equilibrated sequentially in the following solutions: 3 mL of buffer I (Tris-HCl 50 mM pH 8.8, urea 6 M, glycerol 30%, weight/vol, SDS 2% weight/vol, and DTT 10 mg/mL) for 15 min, 3 mL of buffer II (Tris-HCl 50 mM pH 8.8, urea 6 M, glycerol 30%, weight/vol, SDS 2% weight/vol, and iodoacetamide 25 mg/mL) for 15 min, and finally for 15 min in protein running buffer 1X (0.025M Tris, 0.192M glycine, 0.1% SDS). Then, the strips were directly applied to 12% polyacrylamide gels using the SE 600 Ruby Standard Dual Cool Vertical Unit (Amersham Biosciences, GE Healthcare). Electrophoresis was carried out at 5 mA/gel for 20 min, and at 15 mA/gel for 9 h. Following separation in the second dimension, the gels were fixed and then stained with Coomassie brilliant blue G-250 (Bio-Rad, Hercules, CA, USA).

### 2.6. Images Analysis and Protein Identification

Two-dimensional polyacrylamide gel electrophoresis (2D-PAGE) gels were analyzed using the software BioNumerics 2D (v6.6; Applied-Maths) as follows: The representative AIEC OMP 2D-PAGE profile was generated using the “*Matching gel 2D type*” tool, which can be used to combine gel images, using the molecular masses and isoelectric point (pI) values as a reference. This representative 2D-PAGE gel included protein spots present in at least 5/7 (>70%) of the OMP profiles of AIEC strains. Subsequently, to identify the protein spots that were highly frequent in AIEC strains but absent in *E. coli* HS, the AIEC representative gel was superimposed on the OMP profile of the commensal *E. coli* HS. Twenty-four representative protein spots were selected and further identified by matrix-assisted laser desorption ionization–time of flight (MALDI-TOF/TOF) mass spectrometry.

### 2.7. PCR Assay

Bacterial strains were analyzed with PCR to detect genes encoding ChuA, FitA, EefC/NodT, NmpC, IrpC, OmpT, and a potential iron receptor (abbreviated here as Lgc). The primer sequences used for PCR are shown in Table 1. These primers were designed using the NRG857c and 541-1 genomes and the PimerBLAST tool (http://www.ncbi.nlm.nih.gov/tools/primer-blast/; accessed on 1 March 2018). The specificity of each primer was verified by comparison with the genomes available in the GenBank database using BLASTn (http://www.ncbi.nlm.nih.gov; accessed on 1 March 2018).

### 2.8. Gene Screening

The full coding sequences of the target genes were screened in the genomic sequences by using the large-scale blast score ratio (LS-BSR) software [36] with the tblastn option. A strain was considered positive for a target gene when BSR ≥ 0.9.

### 2.9. Gene Markers and Classifier Validation

A binary logistic regression analysis was performed to find a predictive model for AIEC identification based on the *chuA*, *eefC*, and *fitA* genes. The performance of the *chuA*, *eefC*, and *fitA* genes as molecular markers of AIEC was also evaluated by implementing the random forest (RF) and naive Bayes (NB) classification algorithms in the Orange data mining suite, V.3.27.0 http://orange.biolab.si (accessed on 1 April 2021). RF is an ensemble method using decision trees and NB is a generative model. A total of 66% of the 81 strains that were used in the phylogenetic analysis were used as the training set and the rest were used as the test set with the five-fold cross-validation method. The results of the cross-validation (classification accuracy, sensitivity, and specificity) were registered and depicted with ROC curves.

### 2.10. Statistical Analysis

A statistical correlation analysis between genes and specific groups of strains was performed using contingency tables with odds ratios. The statistical significance of these associations was determined with Pearson’s chi-square test or Fisher’s exact test (when frequencies were less than 5). When any of the cell values in the contingency tables was zero, Haldane correction was used by adding 0.5 to all cells [37].

## 3. Results

### 3.1. Identification of AIEC Strains Isolated from Patients with Crohn’s Disease

A set of 16 *E. coli* strains isolated from patients with Crohn’s disease were included in this study (Appendix A). These strains were analyzed by PFGE to determine their genetic relationships. The dendrogram obtained showed a wide variety of PFGE patterns with a genetic similarity coefficient ranging from 69% to 95%, which were grouped into two major clusters (Figure 1A, clusters I and II). Phylogenetic groups B2 (8/16 strains) and D (7/16 strains) were the most frequent; however, no apparent correlation was found between the PFGE pattern clustering and the phylogenetic grouping of the strains. Of note, 5/16 strains were resistant to ampicillin, 5/16 were multi-resistant, and 6/16 were sensitive to the eight antibiotics tested. Collectively, these results indicate that these strains comprise a heterogeneous group of *E. coli*.

Currently, AIEC strains are identified based on their abilities to adhere to and invade intestinal epithelial cells and to survive and replicate within macrophages. Therefore, we next investigated these virulence phenotypes in the strains. For this, the non-adherent and non-invasive *E. coli* HB101 strain was used as a negative control, while the AIEC reference HM605 and NRG857c strains were used as positive controls. The cut-off values used in these in vitro assays are indicated in the Materials and Methods section. As a result, 14 strains were considered adherent while only two strains (1I06 and 7C02) were non-adherent (Figure 1B). In addition, 15 strains showed the ability to invade Caco-2 cells, while the 1I06 strain did not (Figure 1C). Regarding the ability to survive and replicate within macrophages, seven strains (4C01, 4I01, 5I01, 6I09, 9C01, 10I01, and 18I08) had a survival value equal to or greater than that exhibited by the reference HM605 and NRG857c strains (Table 2). Taken together, these results indicate that only seven strains (4C01, 4I01, 5I01, 6I09, 9C01, 10I01, and 18I08) exhibited the virulence phenotypes required to be defined as putative AIEC (Appendix A).

### 3.2. Outer Membrane Protein Profiles of Putative AIEC Strains

In the post-genome era, research into proteomics provides the opportunity to analyze thousands of proteins in complex mixtures. Combining the 2D-PAGE and mass spectrometry techniques has become a powerful methodology that allows the identification of protein spots of interest [38]. We implemented this proteomic approach to obtain the OMP profiles of the seven strains defined as AIEC and then compare them with the OMP profile of the commensal *E. coli* HS strain. Through these proteomic analyses, we sought to identify OMPs associated with AIEC. Of note, the strains were grown in DMEM containing bile salts because they are an environmental signal that upregulates genes associated with AIEC colonization [20,33].

Figure 2 shows the AIEC OMP profiles obtained by 2D-PAGE. OMP extracts were resolved in the pH range of 4 to 7, leading to observation of more than 40 protein spots per strain. AIEC OMP profiles were combined in silico to obtain a global AIEC OMP profile, which was then compared with (superimposed on) the HS OMP profile (Figure 2H). After this analysis, 24 protein spots were present in at least 70% of AIEC strains and absent in the HS strain. These protein spots were cut from the gels and analyzed by MALDI-TOF/TOF mass spectrometry. The *e-values* of 2 of the 24 protein spots were not significant (*e-values* of 2.5 and 12.5), due to which they were discarded, whereas 12 protein spots corresponded to 6 proteins (OmpA, FepA, LptD, BtuB, TolC, and Dps), having more than one isoform. As a result, a total of 16 OMPs were identified, most of them with functions related to the uptake of iron and porins (Table 3).

### 3.3. Frequency of Detection and Distribution of the Genes Encoding OMPs of Interest

The 16 OMPs were present in five (71%) to seven (100%) of the AIEC strains characterized in this study (Appendix A). However, the presence of isoforms, aberrant migration of proteins, and the different expression programs between the AIEC and HS strains could have biased our results. Therefore, we performed a BLAST analysis to rule out the presence of the genes encoding these proteins in the genome of the HS strain and two additional commensal *E. coli* strains (IAI1a and SE11 strains). This analysis was also performed on a set of seven reference AIEC strains (LF82, HM605, NR6857, UM146, 541-1, 541-15, 576-1) whose complete genomes are available in GenBank (Appendix A). As a result, 7/16 genes (*chuA*, *eefC*, *fitA*, *irpC*, *lgC*, *nmpC*, and *ompT*) were absent in the HS and IAI1a genomes. Except for the *nmpC* and *ompT* genes, the above genes were also absent in the SE11 genome. In contrast, these seven genes were highly frequent in the AIEC genomes and, consequently, we focused our study on them.

Next, we developed a PCR assay that allows the molecular identification of these seven genes and extended our search for them to a set of commensal *E. coli* strains from our collection (fecal isolates from healthy individuals). Consistent with the previous findings, except for the *irpC* and *nmpC* genes, the other five genes were rare in commensal strains, with prevalence below 13% (Appendix A).

To further assess the distribution of these genes, we analyzed 81 genomes from strains of several origins, including AIEC, non-AIEC from patients with Crohn’s disease, commensals, DEC, and ExPEC. According to a core SNP-based phylogenetic tree (Figure 3A), *chuA* was the most widely distributed gene among AIEC, being harbored by strains belonging to phylogroups B2, G, and D, followed by the *irpC* gene, which was identified among AIEC from phylogroups B2, D, and B1. However, both genes were also identified among some non-AIEC strains isolated from patients with Crohn’s. By contrast, the *eefC* and *fitA* genes showed a more limited distribution, being identified almost exclusively in AIEC strains of the phylogroup B2. The *lgC* gene was detected in a small group of AIEC of the phylogroup B2. Regarding the *nmpC* and *ompT* genes, they were widespread among strains of different origins. The frequency of detection of each gene among AIEC and other strains is shown in Figure 3B.

Subsequently, the correlation between each gene and specific groups (AIEC, Commensal, Non-AIEC, and DEC) of strains was determined. Notably, the *chuA*, *eefC*, and *fitA* genes were significantly correlated with AIEC when compared to strains from other groups (Figure 3C). The *irpC* gene was significantly correlated with AIEC but also with the non-AIEC strains isolated from patients with Crohn’s disease. On the other hand, the *lgc* gene was found with low frequency among the strains analyzed, lacking statistical correlation with any group of strains. Similarly, the *nmpC* gene was distributed in strains from different phylogenetic origins without being significantly correlated with any group. The *ompT* gene was found with a high frequency among AIEC but also in DEC, being significantly correlated with the latter group of strains. Taken together, these results indicated that the *chuA*, *fitA*, *and eefC* genes are widespread and correlated with AIEC strains.

We next sought to investigate the usefulness of the *chuA*, *fitA*, *and eefC* genes as molecular markers of AIEC. To do this, the presence/absence of the *chuA*, *fitA*, and *eefC* genes in the 81 genomes was analyzed using a binary logistic regression. As result, it was found that these three genes enabled AIEC to be distinguished from other *E. coli* strains with an accuracy of 81.5% (sensitivity 77.8%, specificity 82.5%) (Appendix A). We also implemented the random forest (RF) and naive Bayes (NB) classification methods and evaluated the performance of these genes. By implementing the RF algorithm, the combination of *chuA + fitA + eefC* enabled AIEC to be distinguished from other *E. coli* strains with an area under the curve (AUC) and accuracy of 0.82 and 81%, respectively, while with the NB algorithm an AUC and accuracy of 0.81 and 84% were achieved, respectively (Figure 3D,E, Appendix A). Thus, these results highlight the potential use of these three genes in the molecular characterization of AIEC strains.

## 4. Discussion

Numerous studies have documented a higher detection frequency of AIEC strains in patients with Crohn’s disease compared to controls, which has supported the correlation of these bacteria with the etiopathogenesis of CD. Despite this, the specific role of AIEC in the onset, progression, and/or reactivation of CD has not yet been established [39]. An important limitation in deciphering this is the difficulty in identifying AIEC strains. Until now, the identification of these strains has been based on time-consuming in vitro assays and culture techniques, which are hard to standardize. Since treatments for CD are limited and non-healing, it is of the utmost importance to know the pathogenic mechanisms of the AIEC strains, as well as to develop molecular tools for their rapid identification. Moreover, an easy detection of AIEC strains could facilitate epidemiological studies investigate possible environmental and animal reservoirs and transmission pathways. Ultimately, the identification of AIEC carriers could be useful for CD prevention, allowing personalized therapies to be applied [16].

In this work, we focused on the study of a collection of AIEC strains to identify specific markers that could be useful in the molecular detection of these pathogens. To achieve this objective, we first phenotypically characterized a collection of 16 *E. coli* strains isolated from CD patients. In particular, we determined that seven of these strains exhibited the pathogenic phenotypes required to be classified as putative AIEC (Figure 1B,C, Table 2). The remaining nine strains failed to reach the cut-off values for adherence, invasiveness, and survival within macrophages characteristic of AIEC, which were defined on the basis of previous studies [2,4,31]. However, we consider that these “non-AIEC” strains cannot be classified merely as commensal *E. coli* because they were isolated from the intracellular content of biopsied tissues from CD patients (see the Materials and Methods section). Therefore, these strains must have some degree of invasiveness or ability to survive within human cells. In fact, it could be that the in vitro phenotypic characterization failed to fully demonstrate the pathogenicity that these strains may display in the human intestine. This represents a clear example of the current difficulty and arbitrariness of defining and differentiating a putative AIEC from other *E. coli*.

Once the seven putative AIEC strains were selected, we implemented an approach combining proteomics and genomics analyses. Specifically, the OMP profiles of these strains were characterized and subsequently compared with the OMP profile of the HS strain (Figure 2). It is important to mention that the strains were grown in a culture medium supplemented with bile salts, which are an environment signal that positively regulates the expression of genes associated with colonization in AIEC [20,33]. In this way, we tried to induce the expression of genes that AIEC may require during its stay in the human intestine.

Interestingly, we were able to identify a set of 16 proteins that were present in most of the OMP profiles of the AIEC strains but not in the HS strain (Table 3). However, a further genomic analysis showed that only 7/16 genes (*chuA*, *eefC*, *fitA*, *irpC*, *lgC*, *nmpC*, and *ompT*) coding for these proteins were absent in the HS strain (Appendix A). We hypothesize that the 9/16 genes (*cirA*, *fepA*, *lptD*, *tolC*, *btuB*, *tufA*, *dps*, and *ompA*) detected in the HS genome were not expressed by this bacterium under the culture conditions used. Alternatively, it is possible that we could not identify these proteins in the 2D-PAGE gel of the HS strain due to variations in electrophoretic migration. The search for these genes was also performed in seven AIEC reference strains and two reference commensal *E. coli* strains (IAI1 and SE11), the complete genomes of which are available in GenBank. After this analysis, it was clear that the *chuA*, *eefC*, *fitA*, *irpC*, *lgC*, *nmpC*, and *ompT* genes are very frequent in the AIEC genomes (Appendix A). Collectively, these results led us to determine the frequency of these seven genes in a larger collection of strains.

Considering that there are few genomes of commensal *E. coli* strains available in databases, we determined with PCR the frequency of these genes in a collection of *E. coli* strains (fecal isolates from healthy individuals) available in the laboratory. This analysis revealed that, except for the *irpC* and *nmpC* genes, the other five genes are infrequent in commensal *E. coli* strains (Appendix A). Interestingly, 6/7 of these genes were detected in one (strain 3) of these strains. In this sense, it has been reported that ExPEC strains have the intestinal microbiota as a reservoir, with a prevalence reported in fecal isolates of around 10% [40]. In line with this, we do not rule out the possibility that this strain is actually an ExPEC.

We also investigated the phylogenetic distribution of these genes and found that they are not characteristic of all AIEC; however, they are associated with a particular group of these strains, which are phylogenetically related and belong to phylogroup B2 (Figure 3A). It should be emphasized that *E. coli* strains belonging to phylogroup B2 are highly prevalent in individuals with intestinal inflammatory diseases [41] and are also associated with greater virulence in the ExPEC strains [42,43].

It is interesting to note, however, that these genes were also identified in a number of phylogenetic related non-AIEC strains (isolated from CD patients) belonging to phylogroups B2, F, and D. This raises the question: can the detection of these genes highlight the pathogenic potential of these non-AIEC strains? As previously mentioned, these strains were classified as non-AIEC because they did not exhibit the characteristic phenotypes of putative AIEC. However, these in vitro tests are quite arbitrary, and they are not necessarily intended to demonstrate the harmless nature of these strains. In future research, comparative transcriptomic studies can be carried out between AIEC and non-AIEC strains that contain these genes. The results of such investigations could reveal a possible association between the expression of these genes and the pathogenicity of AIEC.

Statistical correlation analysis was used to determine the strength of the association between the genes and specific groups (AIEC, commensal, non-AIEC, and DEC) of strains. As expected, this analysis revealed that the *chuA*, *eefC*, *fitA*, and *irpC* genes are significantly correlated with AIEC (Figure 3C). However, the *irpC* gene was also significantly correlated with non-AIEC strains. Conversely, a negative correlation was found between the *chuA*, *eefC*, *fitA*, and *irpC* genes and DEC strains.

In the case of the *ompT* gene, it was significantly correlated with DEC strains (Figure 3C), which is consistent with previous studies [26]. Due to its high frequency of detection among DEC strains, this gene was not statistically correlated with AIEC. However, it is noteworthy that the frequency of *ompT* in AIEC was 77.8% (Figure 3B). Since different variants of *ompT* are present in pathogenic *E. coli* strains [44], it would be valuable to investigate the function of this gene among AIEC strains.

An important finding of this study that deserves emphasis was that the *chuA*, *fitA*, and *eefC* genes have a potential use as markers of AIEC. The combination of *chuA* + *fitA* + *eefC* allowed AIEC to be distinguished from other *E. coli* strains with a good discriminatory power (accuracy of 81.5%, sensitivity of 77.8%, and specificity of 82.5%) (Appendix A). In a previous study, the *chuA* gene was the most significant individual marker of AIEC, having an accuracy of 56% (93% sensitivity and 41% specificity) [23]. However, cross-validation of these three genes in diverse AIEC collections from different geographical regions will confirm their potential use as markers.

Other genes that have been proposed as molecular markers of AIEC include *pduC* (accuracy of 65%, sensitivity of 50%, and specificity of 80%) [14], *lpfA* (accuracy of 75%, sensitivity of 71%, and specificity of 80%) [14], *lpfA* + *gipA* (accuracy of 83%, sensitivity of 31%, and specificity of 100%) [45], and *pic* + AmpR (accuracy of 75%, sensitivity of 86%, and specificity of 67%) [46]. Therefore, to date, there is no exclusive molecular marker for AIEC, although these genes can be used to complement the molecular characterization of these strains. Moreover, an important limitation of this study and those mentioned previously is the low number of strains analyzed and the need for cross-validation in other collections of AIEC and non-AIEC strains.

In addition to the potential use of these genes as markers of AIEC, there are other reasons they are an interesting target to study, mainly concerning their possible relation with the pathogenicity of this pathotype. On the one hand, it has been seen that the *chuA* gene is widely distributed among pathogenic *E. coli* strains and shares a high homology with the *shuA* gene described in *Shigella dysenteriae* type 1. This gene codes for an OMP responsible for the uptake of iron heme complexes [47,48], and its possible participation in pathogenicity has been evaluated in several *E. coli* strains. It has been reported that it contributes to the renal infection produced by Uropathogenic *E. coli* strains in a murine model of pyelonephritis [49]; however, it seems not to be essential for the invasion or multiplication of *S. dysenteriae* in Henle cells [47], nor does it contribute to bacteriemia of NMEC strains [50], suggesting its presence is only related to affording these strains an adaptive advantage. In the case of AIEC strains, it has been seen that there is a correlation between the presence of ChuA and the ability to survive in macrophages, which is why its possible participation in such a process has been suggested [14].

On the other hand, for the EefC/NodT (outer membrane channel) and FitA (potential iron-ferrichrome receptor), information is more limited. The outer membrane channel has a 100% identity with a channel in the NodT family and with EefC. NodT, as described in *Rhizobium leguminosarum*, is related to a family of OMPs that include TolC, PrtF, CyaE, and AprF, involved in protein secretion [51]. In contrast, EefC has been described as part of an effluent system associated with quinolone resistance [52]. No reported data could be found for FitA, except that it possesses protein domains corresponding to an iron ferrichrome receptor. Therefore, in both cases, assays are needed to clarify their function and then to assess if this has any relation to the characteristics inherent to AIEC strains.

In conclusion, our results strongly suggest the potential use of the *chuA* + *fitA* + *eefC* genes as markers of AIEC. We consider that the identification of these genes can contribute to the characterization of *E. coli* strains isolated from CD patients, as well as complement the current phenotypic identification of these pathogens.

## Figures and Tables

**Figure 1 ijms-23-09005-f001:**
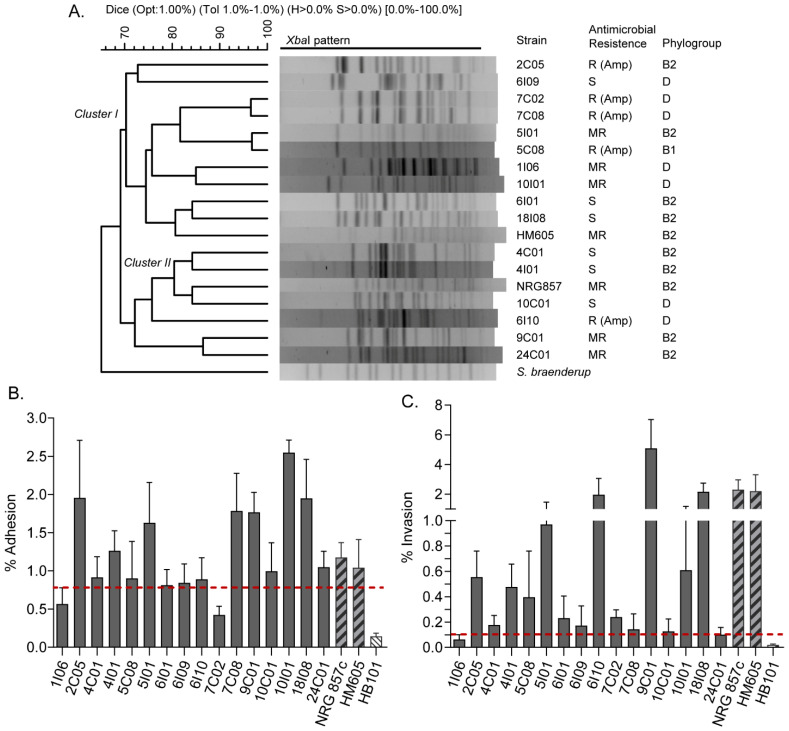
PFGE analysis and phenotypic characterization of 16 *E. coli* strains isolated from patients with Crohn’s disease included in this study. (**A**) Dendrogram generated with *Xba*I-digested PFGE patterns. Phylogenetic groups, antibiotic resistance and the two main clusters (I and II) are indicated. AIEC reference strains HM605 and NRG857c are also included. Strain resistance to eight antibiotics (gentamicin, ciprofloxacin, chloramphenicol, cefotaxime, ampicillin, cefepime, trimethoprim-sulfamethoxazole, and nalidixic acid) was evaluated. S: sensitive; A: resistant; MR: multi-resistant (strain resistant to more than one antibiotic). (**B**) Bacterial adhesion to Caco-2 cells. Error bars represent SD (*n* = 3). (**C**) Bacterial invasion to Caco-2 cells. Error bars represent s.d. (*n* = 3). The dotted red line indicates the cut-off value to consider the strains as adherent or invasive (see Materials and Methods section).

**Figure 2 ijms-23-09005-f002:**
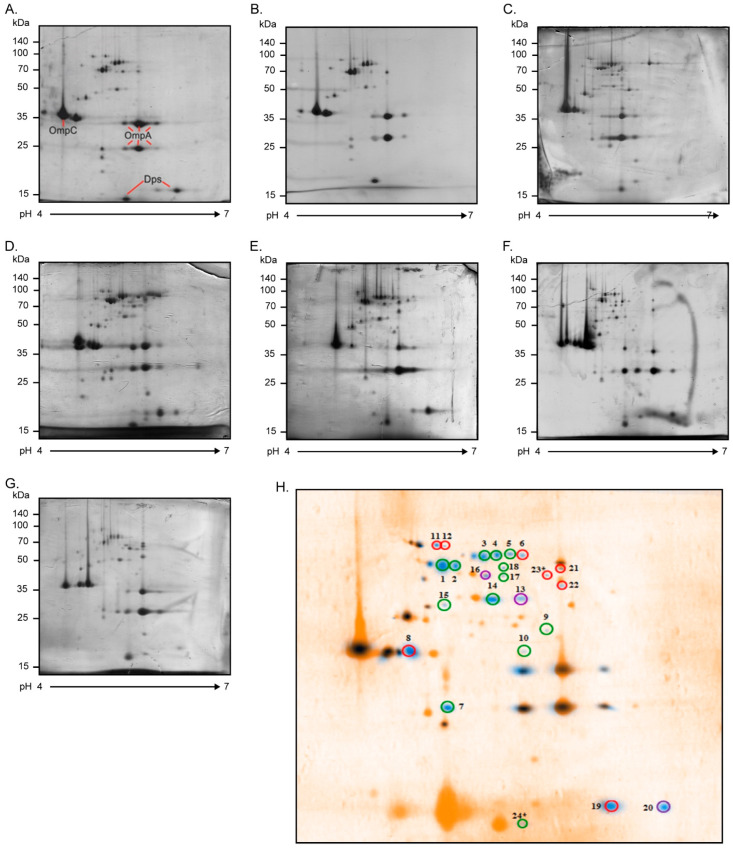
OMP profiles of AIEC strains. Two-dimensional polyacrylamide gel electrophoresis (2D-PAGE) OMP profiles of the AIEC strains grown in DMEM (2% bile salts) at 37 °C overnight: (**A**) 4I01, (**B**) 4C01, (**C**) 5I01, (**D**) 6I09, (**E**) 9C01, (**F**) 10I01, and (**G**) 18I08. Twelve percent polyacrylamide gels (13 cm; pH range: 4–7) were stained with Coomassie blue G-250. The scale bars on the left indicate molecular weights in kDa. Proteins with a well-known 2D-PAGE pattern are indicated (OmpA, OmpC, and Dps). (**H**) A representative AIEC OMP profile (blue spots) involving all the proteins present in at least 70% of the strains was superimposed on the commensal *E. coli* HS protein profile (orange spots). This analysis was performed using BioNumerics 2D software (v6.6). A total of 24 protein spots expressed differentially were selected and identified with MALDI-TOF/TOF mass spectrometry. These spots were absent in the HS OMP profile and highly frequent in the AIEC OMP profiles, as follows: 7/7 (100%, enclosed in green), 6/7 (86%, enclosed in purple), and 5/7 (71%, enclosed in red). * Spots 23 and 24 had no significant e-values in the MALDI-TOF/TOF analysis and were therefore discarded.

**Figure 3 ijms-23-09005-f003:**
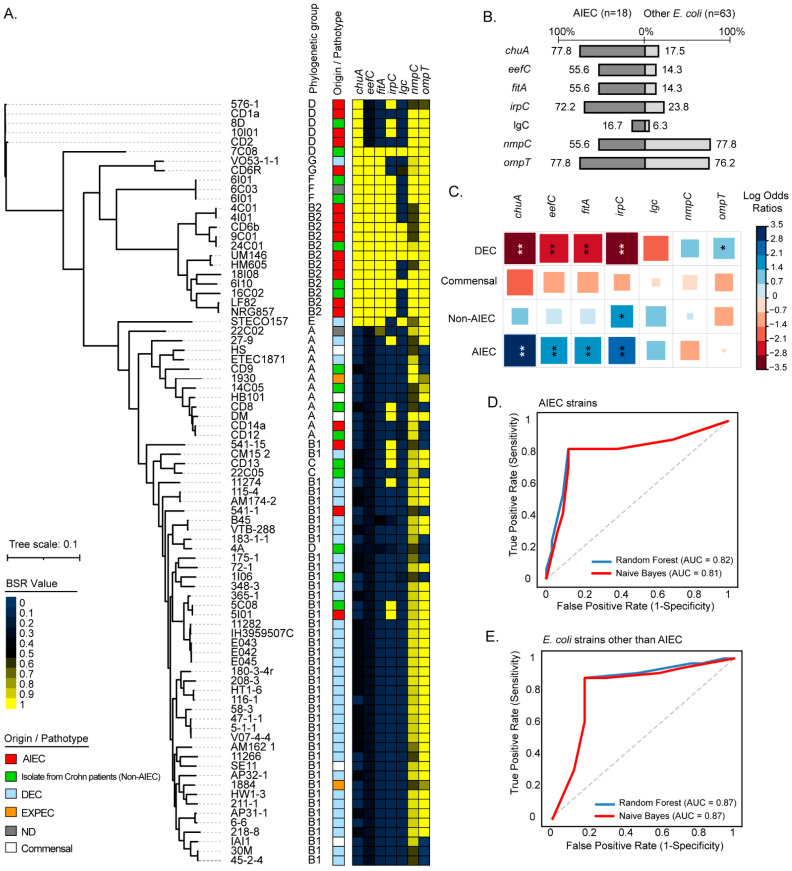
Phylogenetic distribution and prevalence of the *chuA*, *eefC*, *fitA*, *irpC*, *lgC*, *nmpC*, and *ompT* genes among AIEC strains. (**A**) The maximum parsimony phylogenetic tree (midpoint rooted) was based on 7515 core SNPs identified in 81 complete or draft genomic sequences of *E. coli* strains. Phylogenetic groups are indicated. The origin/pathotype of each isolate is indicated by colored boxes according to the legend. The presence, absence, and variation of each gene are indicated by the color scale, which represents the blast score ratio (BSR) values. A value of 1 indicates the presence of a gene encoding an identical protein. (**B**) Prevalence of the genes in AIEC and other *E. coli* strains. Numbers indicate gene prevalence in percentages in relation to the total strains in each group. (**C**) Pairwise association plot for pathotypes and the indicated genes. Red and blue squares represent negative and positive associations, respectively. The color scale represents the magnitude of the association determined by odds ratios. * *p* < 0.05, ** *p* < 0.005 determined by Pearson’s chi-square test or Fisher’s exact test. (**D**,**E**) Evaluation of the *chuA*, *fitA*, and *eefC* genes as molecular markers to discriminate AIEC strains from other E. coli strains. The random forest and naive Bayes classification methods were implemented. The performance of each classifier model is represented by ROC curves and the area under the curve (AUC).

**Table 1 ijms-23-09005-t001:** Primer sequences used in this study.

Gene	Sequence (5′-3′)	Tm (°C)	Product Length (pb)	Control (+)	Reference
*chuA*	GACGAACCAACGGTCAGGAT TGCCGCCAGTACCAAAGACA	59	279	NRG 857c	[35]
*fitA*	ATCCCGCAGGTGGTCAATAC GGCATCGGCAACGAAATAGC	60	1758	NRG 857c	This study
*eefC/NodT*	TTGAGTGCGGGATGTGTCTC CCGTCAACACGGTCAGGTAA	60	1225	NRG 857c	This study
*nmpC*	ACTGATGGCGATGTCTGCTC ACGCACCCAAGTCTTTTCCT	62	863	NRG 857c	This study
*irpC*	ATCAGCCAACAACGTCTCGT GCGTATCAATCACGCCGTTC	60	1622	NRG 857c	This study
*ompT*	GCCTGCACCATTTTTGCTGT TCCTGACAACCCCTATTGCG	59	878	NRG 857c	This study
*lgc*	ACAGCAGGCTGGGAAGAATC TGCTGTACGGTAGCGTTTGT	60	1539	541-1	This study

**Table 2 ijms-23-09005-t002:** Ability of *E. coli* strains to survive within RAW 264.7 macrophages *.

Strains	Initial Uptake (3 h.p.i)		% 24 h.p.i	
1I06	292.0	±78.0	6.9	±0.5
2C05	143.0	±11.9	3.3	±1.0
4C01	679.8	±108.5	409.8	±71.7
4I01	606.8	±104.5	127.3	±21.1
5C08	124.3	±15.2	16.3	±4.4
5I01	168.7	±10.5	83.8	±16.6
6I01	45.1	±39.1	5.1	±4.1
6I09	110.7	±9.9	27.0	±10.6
6I10	8.4	±1.0	1.0	±0.3
7C02	737.0	±148.6	5.7	±1.3
7C08	128.7	±23.8	3.0	±1.9
9C01	250.7	±44.1	59.8	±12.4
10C01	50.6	±4.6	9.0	±3.9
10I01	88.7	±6.2	30.0	±7.0
18I08	83.2	±16.8	45.1	±7.7
24C01	119.2	±9.5	0.003	±0.004
NRG 857c	74.3	±5.2	31.3	±5.2
HM605	58.3	±9.7	22.9	±4.4
HB101	29.7	±7.1	0.0	±0.0

* The table shows the initial uptake, defined as the mean CFUs resisting amikacin treatment after 3 h.p.i, and the percentage of bacterial content at 24 h.p.i compared to the initial uptake. The HB101 strain was used as a negative control. Those strains that survived within macrophages according to the criteria used as reference (equal to or greater than the values exhibited by the AIEC NRG857c and HM605 strains) are highlighted in gray. Mean and standard deviation indicated correspond to three independent experiments performed in duplicate.

**Table 3 ijms-23-09005-t003:** Proteins identified by MALDI-TOF/TOF corresponding to those *spots* present in >70% of the AIEC strains and absent in the commensal *E. coli* HS strain.

Function	No. Spot	Protein	Description	*E-Value*
Iron uptake	1	ChuA	Hemo/iron group receptor	1.58114 × 10^92^
	2	CirA	Siderophore, colicin, microcin receptor	1.25594 × 10^36^
	3	FitA	Iron-ferrichrome receptor	1.25594 × 10^79^
	4,5	FepA	Ferrienterobactin receptor	5 × 10^50^ 6.29463 × 10^55^
	6	Lgc *	Ligand-regulated channel (potential iron receptor)	3.97164 × 10^21^
	18	IrpC	Yersiniabactin/pesticin receptor	1.58114 × 10^54^ 3.15479 × 10^36^
Porins	10	OmpT	Protease, outer membrane protein	6.29463 × 10^44^
	8	NmpC	Outer membrane protein	3.15479 × 10^77^
	21,22	OmpA	Outer membrane protein	7.92447 × 10^61^ 1.25594 × 10^67^
	13	EefC/NodT	Outer membrane channel	3.97164 × 10^56^
	14,15	TolC	Outer membrane protein	3.15479 × 10^20^
Specific channel	16,17	BtuB	B12 vitamin transporter	3.15479 × 10^17^
Stress response	19,20	Dps	DNA-protecting protein	9.97631 × 10^72^ 8.34568 × 10^60^
Others	11,12	LptD	LPS-assembly protein	9.97631 × 10^53^ 3.15479 × 10^5^
	7	Tsx	Nucleoside receptor, phage T6 and colicin K receptor	1.58114 × 10^28^
	9	EF-Tu	Elongation factor Tu	3.15479 × 10^23^

* Abbreviation proposed in this work to refer to a certain protein.

## Data Availability

All the data used in this study were drawn from public databases and were permitted for use.

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
