# Peer review of "Characterization of Adherent-Invasive Escherichia coli (AIEC) Outer Membrane Proteins Provides Potential Molecular Markers to Screen Putative AIEC Strains"

_ijms, 2022, doi:10.3390/ijms23169005_

Round 1

Reviewer 1 Report

In this study, Saitz et al phenotypically characterize 7 new AIEC strains isolated from spanish CD patients. They compare outer membrane proteins expressed in these 7 AIEC strains to those expressed in HS commensal strain and identify 7 OMP expressed in AIEC but not in HS (absence of the genes has been PCR-checked in HS but also in 2 additional commensal strains genomes available in the literature as well as in 8 strains from healthy subjects). They next blast the sequence of these 7 OMPs against 81 E. coli genomes  and show that chuA, fitA and eefC genes correlate well with the AIEC status. The combination of these three genes could help to identify AIEC strains.

Major comments :

- This paper uses a high number of E.coli strains. Some of them are described in Tables or supplemental tables or in the literature, so that it would be useful for the reader to have a summary table mentioning at least the phylogroup, the origin of isolation as well as their AIEC status with the reference of its characterization.

- Line 136-140, please clarify the origin of the strains : Table S2 mentions 38 published strains, where are the 43 other strains from? Are they all characterized for adherent-invasive status ? The project number seems to be unknown on NCBI database at this time.

- What exactly can be considered as a ‘commensal’ strain ? Being isolated from a healthy subject does not seem sufficient given that AIEC strains are regularly identified in such people. Are the « commensal » strains used here characterized for their adherent/invasive phenotype ?

- Maybe a naive question : how do the authors design PCR primers to avoid false negative due to mismatch in genomic sequence of bacterial strains ? Unless I’m mistaken, nothing has been explained about that.

- The authors use NRG857c and HM605 strains as AIEC reference strains, however their survival rate in macrophages seems low (around 30%) at 24h. Is this consistent with what has been shown in the literature ? In the paper from Glasser et al, LF82 AIEC strain seems to replicate far better.

- In Table S5, the authors search for the presence of their set of 7 genes in a few AIEC and commensal strains and express the results as % of strains possessing the genes. This does not seems very significant as it is calculated from a very low number of strains (only 3 commensal strains).

- The adhesion/invasion protocol used by the authors uses amikacin instead of the more classically used gentamycin, this impairs in some extent the comparison with other studies, why did the authors chose this antibiotic ?

- ChuA alone has high sensitivity but low specificity, based on published data. Please clarify sensitivity and specificity in the text and discuss the interest of this 7-genes set in comparison with other molecular markers that have been proposed in the literature : is it a real improvement for AIEC prediction ?

Minor comments :

- Table S5 : the table headers are not readable.

Author Response

Responses to reviewers of Saitz et al.: "Characterization of Adherent-invasive Escherichia coli (AIEC) outer membrane proteins provides potential molecular markers to screen putative AIEC strains".

Manuscript ID: ijms-1841163

We want to thank the reviewers for the insightful comments. The authors fully agreed with their comments and felt that their suggestions and recommendations have greatly increased the quality of the manuscript. Our specific responses to the reviewers’ comments are listed below.

Reviewer 1

Major comments:

- This paper uses a high number of E. coli strains. Some of them are described in Tables or supplemental tables or in the literature, so that it would be useful for the reader to have a summary table mentioning at least the phylogroup, the origin of isolation as well as their AIEC status with the reference of its characterization.

R/ We agree. We thank the reviewer for this suggestion. We have modified Table S1 by adding information on the phylogroup and phenotypic characterization of the strains, indicating which were defined as AIEC.

Similarly, we modified Table S2 to include the full list of genomes used for phylogenetic analysis. Information on the phylogroup, origin, pathotype, and presence/absence of the genes of interest was also added to this table.

- Line 136-140, please clarify the origin of the strains: Table S2 mentions 38 published strains, where are the 43 other strains from? Are they all characterized for adherent-invasive status? The project number seems to be unknown on NCBI database at this time.

R/ We modified Table S2 to include the full list of genomes used for phylogenetic analysis. Information on the phylogroup, origin, pathotype, and presence/absence of the genes of interest was also added to this table.

- What exactly can be considered as a ‘commensal’ strain? Being isolated from a healthy subject does not seem sufficient given that AIEC strains are regularly identified in such people. Are the « commensal » strains used here characterized for their adherent/invasive phenotype?

R/ The E. coli strains that we define as intestinal commensals were characterized in a previous study by Montero et al., 2014. In the aforementioned study, it was determined that these strains lack the characteristic genes of diarrheagenic E. coli pathotypes. This information was added to the manuscript in section 2.1 (lines 128 – 130). However, these strains have not been characterized for their adherence and invasion phenotype.

  • Montero D, Orellana P, Gutiérrez D, Araya D, Salazar JC, Prado V, Oñate A, Del Canto F, Vidal R. Immunoproteomic analysis to identify Shiga toxin-producing Escherichia coli outer membrane proteins expressed during human infection. Infect Immun (2014) 82:4767–77. doi: 10.1128/IAI.02030-14

- Maybe a naive question: how do the authors design PCR primers to avoid false negative due to mismatch in genomic sequence of bacterial strains? Unless I’m mistaken, nothing has been explained about that.

R/ The primers were designed using the NRG857c and 541-1 genomes and the PimerBLAST tool. (http://www.ncbi.nlm.nih.gov/tools/primer-blast/). The specificity of each primer was verified by comparison with the genomes available in the GenBank database using the BLASTn (http://www.ncbi.nlm.nih.gov). This information was added to the manuscript in section 2.7 (lines 253-257).

- The authors use NRG857c and HM605 strains as AIEC reference strains, however their survival rate in macrophages seems low (around 30%) at 24h. Is this consistent with what has been shown in the literature? In the paper from Glasser et al, LF82 AIEC strain seems to replicate far better.

R/ The J774-A1 line was derived from reticulum cell sarcoma (1), while the RAW264.7 line was derived from an Abelson murine leukemia virus-induced tumor (2), both in BALB/c mice. The main difference between the two cell lines is that RAW264.7 macrophages express more iNOS than J774-A1 macrophages (3), which is important for controlling intracellular bacterial pathogens (4). Moreover, although they are genetically similar, AIEC reference strains LF82 and NRG857c (phylogroup B2 and serotype O83:H1) differ in plasmid content and antibiotic resistance genes detected (5, 6). Conversely, the strain HM605 (B2, O1:H7), also considered a reference AIEC strain, is phylogenetically distant from NRG857c and LF82 (7). The observed differences between our results and those obtained by Glasser et al. could be influenced by all these factors; however, it should be noted that both NRG857c and HM605 survive for 24 hours in RAW264.7 macrophages.

  • Hirst, J. W., Jones, G. G., and Cohn, M. (1971). Characterization of a BALB/c myeloma library. J. Immunol. 107, 926–927.
  • Ralph, P., and Nakoinz, I. (1977). Antibody-dependent killing of erythrocyte and tumor targets by macrophage-related cell lines: enhancement by PPD and LPS. J. Immunol. 119, 950–954.
  • Cabral GRA, Wang ZT, Sibley LD, DaMatta RA. Inhibition of Nitric Oxide Production in Activated Macrophages Caused by Toxoplasma gondii Infection Occurs by Distinct Mechanisms in Different Mouse Macrophage Cell Lines. Front Microbiol. 2018 Aug 20;9:1936. doi: 10.3389/fmicb.2018.01936. PMID: 30177926; PMCID: PMC6109688.
  • Chakravortty D, Hensel M. Inducible nitric oxide synthase and control of intracellular bacterial pathogens. Microbes Infect. 2003 Jun;5(7):621-7. doi: 10.1016/s1286-4579(03)00096-0. PMID: 12787738
  • Miquel, S., Peyretaillade, E., Claret, L., De Vallee, A., Dossat, C., Vacherie, B., et al.(2010). Complete genome sequence of Crohn’s disease-associated adherent-invasive coli strain LF82. PLoS One 5:e12714. doi: 10.1371/journal.pone.0012714
  • Nash, J. H., Villegas, A., Kropinski, A. M., Aguilar-Valenzuela, R., Konczy,P., Mascarenhas, M., et al. (2010). Genome sequence of adherent-invasive Escherichia coli and comparative genomic analysis with other coli pathotypes.BMC Genomics 11:667. doi: 10.1186/1471-2164-11-667.
  • Clarke DJ, Chaudhuri RR, Martin HM, Campbell BJ, Rhodes JM, Constantinidou C, Pallen MJ, Loman NJ, Cunningham AF, Browning DF, Henderson IR. Complete genome sequence of the Crohn's disease-associated adherent-invasive Escherichia coli strain HM605. J Bacteriol. 2011 Sep;193(17):4540. doi: 10.1128/JB.05374-11. Epub 2011 Jun 24. PMID: 21705601; PMCID: PMC3165516.

- In Table S5, the authors search for the presence of their set of 7 genes in a few AIEC and commensal strains and express the results as % of strains possessing the genes. This does not seem very significant as it is calculated from a very low number of strains (only 3 commensal strains).

R/ We agree. Table S5 shows the presence/absence of the 16 OMPs of interest in a small number of AIEC and commensal strains that are available in GenBank. This preliminary analysis made it possible to determine the distribution of these 16 genes. Through this analysis, we found that some of these genes were present in the genomes of commensal E. coli strains even though the proteins they encode were not identified in the OMP profile of the HS strain (2D-PAGE analysis).

Therefore, with this analysis we rule out 9/16 genes due to their presence in commensal E. coli strains. Subsequently, we searched for the remaining 7 genes in 81 strains of different origins. This was explained in the manuscript in section 3.3.

This analysis was complemented with the search for these genes by PCR in our collection of commensal E. coli strains.

- The adhesion/invasion protocol used by the authors uses amikacin instead of the more classically used gentamycin, this impairs in some extent the comparison with other studies, why did the authors choose this antibiotic?

R/ To address this issue, we decided to test amikacin (another aminoglycoside) because it had been reported to be effective in this type of assay (x), and all strains used in this study were amikacin sensitive. The results of antibiograms performed on the strains were included (Table S1). It shows that two strains were Gentamicin-resistant, preventing its use in the protection assays.

  • Gupta T, Fine-Coulson K, Karls R, Gauthier D, Quinn F. Internalization of Mycobacterium shottsii and Mycobacterium pseudoshottsii by Acanthamoeba polyphaga. Can J Microbiol. 2013 Aug;59(8):570-6. doi: 10.1139/cjm-2013-0079. PMID: 23899000.
  • Rangel SM, Logan LK, Hauser AR. 2014. The ADP-ribosyltransferase domain of the effector protein ExoS inhibits phagocytosis of Pseudomonas aeruginosa during pneumonia. mBio 5(3):e01080-14. doi:10.1128/mBio.01080-14.

- ChuA alone has high sensitivity but low specificity, based on published data. Please clarify sensitivity and specificity in the text and discuss the interest of this 7-genes set in comparison with other molecular markers that have been proposed in the literature: is it a real improvement for AIEC prediction?

 R/ We thank the reviewer for this suggestion. The accuracy, sensitivity, and specificity of the chuA, fitA, and eefC genes were indicated in lines 399 to 402. Additionally, we compared these genes with AIEC markers proposed in previous studies (lines 516 - 519 and 521 -529).

Minor comments:

- Table S5: the table headers are not readable.

R/ The table was corrected.

Reviewer 2 Report

In this work, various strains of Adherent-invasive E. coli (AIEC) were examined to identify specific markers of pathologies that can be caused by these strains. To do this, the authors compared the levels of adhesion and invasion to colorectal adenocarcinoma Caco-2 cells, 16 strains of AEIC isolated from patients with Crohn's disease. As a result of the work, the genes most correlated with the pathologies caused by AEIC were identified.

Reference 38 is incorrect.

In Fig. 1C, there are no bars for strain 5I01. And, in general, doing such clipped errors in histograms is a bad idea.

Author Response

Responses to reviewers of Saitz et al.: "Characterization of Adherent-invasive Escherichia coli (AIEC) outer membrane proteins provides potential molecular markers to screen putative AIEC strains".

Manuscript ID: ijms-1841163

We want to thank the reviewers for the insightful comments. The authors fully agreed with their comments and felt that their suggestions and recommendations have greatly increased the quality of the manuscript. Our specific responses to the reviewers’ comments are listed below.

Reviewer 2

Reference 38 is incorrect.

R/ After verification, the reference is properly referenced.

In Fig. 1C, there are no bars for strain 5I01. And, in general, doing such clipped errors in histograms is a bad idea.

R/ Figure 1C was corrected.
